# Effects of Dietary Intervention on Nutritional Status in Elderly Individuals with Chronic Kidney Disease

**DOI:** 10.3390/nu16050632

**Published:** 2024-02-24

**Authors:** Nunzia Cacciapuoti, Maria Serena Lonardo, Mariastella Di Lauro, Mariana Di Lorenzo, Laura Aurino, Daniela Pacella, Bruna Guida

**Affiliations:** 1Physiology Nutrition Unit, Department of Clinical Medicine and Surgery, University of Naples Federico II, Via Sergio Pansini 5, 80131 Napoli, Italy; 2Department of Public Health, University of Naples Federico II, Via Sergio Pansini 5, 80131 Napoli, Italy

**Keywords:** CKD, low-protein diet, malnutrition risk, GNRI, geriatric nutritional risk

## Abstract

The prevalence of chronic kidney disease (CKD) is rising, especially in elderly individuals. The overlap between CKD and aging is associated with body composition modification, metabolic abnormalities, and malnutrition. Renal care guidelines suggest treating CKD patient with a low-protein diet according to the renal disease stage. On the other hand, geriatric care guidelines underline the need for a higher protein intake to prevent malnutrition. The challenge remains of how to reconcile a low dietary protein intake with insuring a favorable nutritional status in geriatric CKD populations. Therefore, this study aims to evaluate the effect of a low-protein adequate energy intake (LPAE) diet on nutritional risk and nutritional status among elderly CKD (stage 3–5) patients and then to assess its impact on CKD metabolic abnormalities. To this purpose, 42 subjects [age ≥ 65, CKD stage 3–5 in conservative therapy, and Geriatric Nutritional Risk Index (GNRI) ≥ 98] were recruited and the LPAE diet was prescribed. At baseline and after 6 months of the LPAE diet, the following data were collected: age, sex, biochemical parameters, anthropometric measurements, body composition, and the GNRI. According to their dietary compliance, the subjects were divided into groups: compliant and non-compliant. For the compliant group, the results obtained show no increased malnutrition risk incidence but, rather, an improvement in body composition and metabolic parameters, suggesting that the LPAE diet can provide a safe tool in geriatric CKD patients.

## 1. Introduction

Chronic kidney disease (CKD) is a worldwide public health problem, associated with a high risk of morbidity and mortality, representing a significant public health concern, with increasing incidence and prevalence. It is estimated that 600 million people all over the world suffer from CKD and its prevalence is rising especially among older adults, as a consequence of socio economic development and better life expectancy [1]. Aging and CKD are both associated with metabolic and nutritional derangements, chronic inflammation, malnutrition as well as protein-energy wasting (PEW), important comorbid conditions that predict poor clinical outcomes and an increased risk of multimorbidity, disability and mortality [2,3]. Aging is a natural process affecting skeletal muscle and adipose organ physiology, during which both tissues undergo quantitative and functional changes characterized by an increase in fat mass with a redistribution in favor of visceral depots and a decrease in fat free mass and loss of muscle mass, this last one referred to as ‘sarcopenia’ [4,5,6,7,8,9,10]. On the other hand, CKD itself is associated with functional alterations of both muscle mass and adipose tissue. These derangements are related to a systemic, chronic low-grade inflammation and to an increased risk of metabolic abnormalities, particularly insulin resistance (IR), CKD progression, cardiovascular disease and diabetes [9,10].

Interestingly, IR is a common and very early alteration in elderly CKD patients, having the potential to power up a vicious cycle of impaired muscle function and PEW contributing to the progression of CKD, worsening morbidity and mortality [11,12,13,14]. Therefore, the overlap of body composition changes associated with CKD and aging, metabolic abnormalities, inflammation, IR, and malnutrition represents a very important aspect of the comprehensive management of elderly patients with CKD. The optimal nutritional care for elderly patients with advanced CKD is still uncertain, and there is an urgent need for evidence-based indications regarding the adequate approach in this setting [15]. Indeed, although protein restriction provides direct benefits to CKD patients, it is only a very relevant part of the more complex dietary management of CKD patients, since it is equally important to maintain an adequate energy intake.

The benefits of protein restriction stem from the fact that oral protein loading contributes to glomerular hyperfiltration, accompanying increased hemodynamic stress, increased production of pro-inflammatory cytokines and growth factors, and decreased glomerular membrane selectivity, as well as the vasodilation of the arteriole caused by increased plasma glucagon levels. Thus, hyperfiltration increases intraglomerular pressure and possible protein loss, leading to glomerulosclerosis and tubulo-interstitial fibrosis, which are major contributors to the progression of CKD [16]. In addition, in CKD patients, as the elimination of nitrogen products (derived from protein and amino acid catabolism) gradually decreases, so-called “uremic toxins” accumulate in the blood and tissues, leading to inflammation, anorexia, and nausea, with a consequent decreased energy intake [17]. Therefore, in the nutritional management of CKD patients, it is evidently important to provide an adequate protein and energy intake, to ensure both a proper nitrogen balance and appropriate protein utilization in order to maintain an optimal nutritional status [16].

The high prevalence of malnutrition and sarcopenia in old age has led to guidelines from the European Society for Clinical Nutrition and Metabolism (ESPEN) recommending, for elderly individuals, a daily protein intake at or above 1–1.2 g/Kg of their ideal body weight (IBW)/day, higher than the recommended daily allowances in the general population, presently set at 0.8 g/Kg of IBW/day [18]. Conversely, the strategies for CKD management and its complications include reducing protein intakes. However, the recommended level of dietary protein intake differs across guidelines. The current Kidney Disease: Improving Global Outcomes (KDIGO) guidelines suggest maintaining a dietary protein intake of 0.6–0.8 g/Kg of IBW/day, while the PROT-AGE Study Group recommends a dietary protein intake of 0.8 g/Kg of IBW/day and >0.8 g/Kg of IBW/day for elderly CKD patients with a glomerular filtration rate (GFR) < 30 mL/min and 30 to 60 mL/min, respectively [19,20]. In contrast, the recent 2020 Kidney Disease Outcome Quality Initiative (KDOQI) guidelines on nutrition in CKD recommend moderate-to-severe protein restriction in CKD patients without diabetes (0.4–0.6 g/Kg of IBW/day) in order to slow disease progression, prevent or correct metabolic disorders, maintain an adequate nutritional status, and postpone the start of dialysis [21]. Therefore, both geriatric and nephrology recommendations pose a difficult dilemma with regard to daily protein intake targets to prevent malnutrition and protein energy wasting, to slow CKD progression and improve metabolic abnormalities in CKD elderly patients. On the other hand, both geriatric and renal guidelines underline the need for a concomitant adequate energy intake in order to prevent malnutrition and optimize the anabolic utilization of proteins and muscle mass maintenance. In light of the above data, the goal of this study was to evaluate the effect of a low-protein adequate energy intake (LPAE) diet on nutritional risk and nutritional status among CKD (stage 3–5) elderly patients. Furthermore, a secondary objective was to evaluate the impact of the LPAE diet on metabolic abnormalities associated with CKD (stage 3–5) elderly patients.

## 2. Materials and Methods

### 2.1. Study Design

This is a retrospective monocentric study approved by the Ethical Committee of the Federico II University Medical School of Naples on 18 July 2018 (Project identification code 181/18), and all patients gave written informed consent. Between January 2019 and June 2019, forty-two CKD (3–5 stage) patients with an age ≥ 65 years old attended a dedicated dietary counseling clinic, managed by a medical doctor specialist in clinical nutrition and a dietitian in our department. They were recruited according to the following inclusion criteria: age ≥ 65 years old, CKD (3–5 stage) in conservative therapy, Geriatric Nutritional Risk Index (GNRI) ≥ 98. Individuals with diabetes, bedridden, or with other possible causes of malnutrition (like malabsorptive syndromes, cancer, dementia, depression, neurological disorders, gastrointestinal disorders, or infections) were excluded from the study population. Recruited subjects were advised to consume an LPAE diet, according to KDIGO guidelines, with a protein intake of 0.6–0.8 g/Kg of IBW/day and an energy intake of 30–35 kcal/g/Kg of IBW/day [19,22]. Where needed, protein-free products, made from carbohydrates, almost free of protein, phosphorus, sodium, and potassium, were used to raise energy intakes.

#### 2.1.1. Data Collection

We extracted the following variables from the database at baseline and after 6 months on the LPAE diet: age, sex, biochemical parameters, anthropometric measurements, body composition, Geriatric Nutritional Risk Index (GNRI), and pharmacological treatments. Dietary compliance (with both prescribed energy and protein intake) was estimated after 6 months on the LPAE diet.

#### 2.1.2. Biochemical Parameters

Overnight fasting venous blood samples were gathered from patients, and the levels of serum albumin, serum electrolytes (calcium, phosphorus, potassium), total cholesterol, LDL-cholesterol, HDL-cholesterol (HDL-C), triglycerides (TG), blood glucose, uric acid, blood urea, creatinine, serum hemoglobin, and parathyroid hormone (PTH) were evaluated by using standard analytic laboratory methods. The estimated glomerular filtration rate (eGFR) was evaluated, using CKD-EPI formula [23]. The TG/HDL ratio, a practical alternative to HOMA-IR for identifying subjects with IR, was calculated [24,25]. Blood pressure was measured using an aneroid sphygmomanometer.

#### 2.1.3. Anthropometric Measurements

Subjects had to be without shoes and in light clothes. Body weight and height were determined using a calibrated balance beam scale and a stadiometer (Seca 711; Seca Hamburg, Germany), then body mass index [BMI (Kg)/(m^2^)] was calculated. Waist circumference (WC) was assessed, according to the National Institutes of Health (NIH) protocols, with a no-stretch tape measure, halfway between the lower edge of the rib cage and the iliac crest.

#### 2.1.4. Body Composition Analysis

Body composition was assessed using a bioelectrical impedance analysis (BIA) with an 800 μA current at a single frequency of 50 kHz (BIA 101 RJL, Akern Bioresearch, from Florence, Italy) [26]. The exam was performed according to ESPEN guidelines: the electrodes were placed on the hand and the foot, according to Kushner, while patients lay supine with limbs slightly apart from their body, after an overnight fast [18,27]. The BIA parameters used to assess body composition were fat-free mass (FFM), fat mass (FM) (expressed in %), and phase angle (PA). Skeletal muscle mass (SM) was calculated by using the following BIA equation from Janssen et al. 2000 [28]:SM (Kg) = [(h2/BIA resistance × 0.401) + (gender × 3.825) + (age × 0.071)] + 5.102 
where height (h) is expressed in cm, and BIA resistance in ohms. For gender: men = 1 and women = 0. Age in years is used [28]. This value can be converted into the skeletal muscle mass index (SMI) by dividing the limb skeletal muscle mass (Kg) by the square of the height (m^2^).

#### 2.1.5. Definition of the Geriatric Nutritional Risk Index

The GNRI has been described by Bouillanne et al. and it is an effective and simple risk index that evaluates nutritional risk and a proven predictive index for prognosis in elderly individuals, those undergoing dialysis, cardiovascular patients, and those requiring health care. The GNRI formula is as follows:GNRI = GNRI = [1.489 × serum albumin (g/L)] + [41.7 × (actual weight/ideal weight)]. 

Ideal weight was calculated using the Lorentz equation (for men: H − 100 − [(H − 150)/4]; for women: H − 100 − [(H − 150)/2.5]; H: height). 

Patients were assigned to one of two different malnutrition risk groups referring to the original GNRI classification: no nutritional risk (GNRI ≥ 98) or malnutrition risk (GNRI < 98) [29].

#### 2.1.6. Dietary Intake and Dietary Compliance

Dietary intake was evaluated with the use of food frequency questionnaires (FFQs), performed at baseline and after 6 months of the LPAE diet [30,31,32]. Dietary compliance to the prescribed energy and protein intake was estimated after 6 months on the LPAE diet, using a ratio (Rt) of actual vs. recommended intake ×100%. Fair dietary compliance was defined as energy Rt % ≥ 90% and protein Rt < 110% and poor compliance as energy Rt % < 90% and protein Rt ≥ 110 [33]. 

### 2.2. Endpoints

The primary endpoint was to estimate the nutritional risk (GNRI) score and body composition changes in the compliant and non-compliant groups after 6 months on the LPAE diet.

The secondary endpoint was to evaluate the improvement or worsening of blood metabolic and calcium–phosphorus metabolism parameters associated with CKD stages 3–5, after 6 months on the LPAE diet.

### 2.3. Statistical Analyses

Data were expressed as mean ± standard deviation of the mean (DS). The paired-samples *t*-test, independent-samples *t*-test, and chi-squared test were performed. All statistical analyses were performed using SPSS20 (SPSS Inc., Chicago, IL, USA). The statistical significance was set at *p* < 0.05.

## 3. Results

The baseline demographic characteristics, anthropometric measures, body composition, and metabolic parameters of the study population (n = 42; 83% males; mean age of 71.5 ± 5.5 years old; mean BMI of 27.3 ± 3.1 Kg/m^2^) are detailed in Table 1. After 6 months from the baseline, participants were classified as compliant or non-compliant, according to protein and energy intake (Table 2). The compliant group (n = 19; 45.2%) was adherent to both the prescribed protein and energy intake; the non-compliant group (n = 23; 54.8%) showed a significantly increased protein (*p* < 0.05) and a significantly decreased energy intake (*p* < 0.05) compared to the prescribed ones (Table 2). Table 3 shows the results after 6 months in both compliant and non-compliant patients. At the beginning of the study, the groups did not differ in terms of anthropometric, clinical, or demographic features. After 6 months from the baseline, the GNRI was ≥98 in all compliant patients, while two patients in the non-compliant group with a baseline no-nutritional-risk status worsened towards malnutrition risk (GNRI score 98 vs. 91 and 87, respectively). Furthermore, a significant improvement in FFM (77.8 ± 7.1% vs. 77.1 ± 7.5%, *p* < 0.05), SM (40.5 ± 5.5 Kg vs. 39.2 ± 46 Kg, *p* < 0.05), and SMI (14.6 ± 1.2 Kg/m^2^ vs. 14.2 ± 1.1 Kg/m^2^, *p* < 0.05), and a significant decrease in FM (21.8 ± 7.6% vs. 23.4 ± 8.1%, *p* < 0.05) and WC (101.1 ± 10.3 cm vs. 98.4 ± 9.1 cm, *p* < 0.05) were observed in the compliant group only. Still, a significant increase in systolic blood pressure (136.4 ± 13.0 mmHg vs. 128.9 ± 13.6 mmHg, *p* < 0.05) was detected in the non-compliant group. On the other hand, no significant changes were observed in BMI, albumin, or hemoglobin either in the compliant or in non-compliant groups (Table 3 and Table 4). After 6 months from the baseline, a significant improvement was also observed in the biochemical parameters of plasma glucose, serum urea levels, HDL-C, triglycerides, serum phosphate, PTH levels, serum uric acid levels, and TG/HDL ratio (*p* < 0.05) in the compliant group but not in non-compliant patients, who, on the contrary, showed a significant increase in serum phosphate and PTH levels (*p* < 0.05). Moreover, in both the compliant and non-compliant groups, no significant difference was observed regarding eGFR after 6 months compared to the baseline (Table 4). Remarkably, our data show a percentage decrease in uric acid, phosphate, PTH, and the TG/HDL ratio index in the compliant group (−21.7%, −11.6%, −7.4%, and −18.8%, respectively), and a percentage increase in the non-compliant group (+4.4%, +14.5%, +39%, and +4%, respectively), from the baseline to 6 months, with significant differences between the two groups (*p* < 0.05) (Figure 1). No modifications in the patients’ pharmacological therapy were observed during the observation period for any of the population.

## 4. Discussion

In the literature, there are conflicting opinions about the need for nutritional care, with specific regard to protein intake, in elderly CKD patients. The present study contributes to answering open questions on this topic.

The benefits of a low protein intake in reducing hyperfiltration and slowing the progression of kidney disease are widely described in the literature. This nutritional intervention must, however, be accompanied by an adequate energy intake in order to obtain a correct nitrogen balance and the optimal use of proteins by our body to avoid the loss of muscle mass and to reduce malnutrition risk. Hence, patients’ follow-up and adherence to the prescribed nutritional treatment is crucial.

Our data showed a satisfying compliance rate of 45.2% with the prescribed dietary intake, considering as “compliant” those patients who adhered to both the prescribed protein and energy intake. Similar to our findings, previous studies on dietary restrictions show that adherence is a challenge for many CKD patients, ranging greatly between 20% and 70% [34,35]. However, it also represents a challenge for those involved in the nutritional management of CKD patients and, as with other therapies, jeopardizes the achievement of the set objectives.

The main finding of our study was that an LPAE diet, monitored by a nutrition unit dedicated to care, does not increase malnutrition risk and improves body composition and metabolic parameters in CKD elderly patients.

Malnutrition in older adults can be caused by a variety of factors, including a loss of appetite, a lack of ability to chew and swallow, depression, dementia, chronic diseases, and an increased use of prescription medications [36]. Considering all these factors affecting the risk of malnutrition, the recent ESPEN guidelines suggest that high-protein diets may counterbalance sarcopenia and malnutrition in elderly individuals [18]. We should note that the challenge remains of how to reconcile a low dietary protein intake with ensuring a favorable nutritional status and avoiding protein-energy wasting risks in geriatric CKD populations.

There is no single measure to assess malnutrition in CKD patients. So far, in this study, we have determined the GNRI, anthropometric data, and biochemical parameters of patients and performed bioelectrical impedance analysis to assess their nutritional status and body composition. In the present study, we did not observe an associated increased incidence of malnutrition risk but, rather, an improvement in body composition after 6 months on the LPAE diet.

Contrasting data are reported in the literature regarding the effect of a low-protein diet on nutritional status in CKD patients. In 2022, Caldiroli et al. observed that, in older CKD patients at risk of malnutrition, the prescription of a low-protein diet does not induce malnutrition, as detected using the malnutrition inflammation score [37]. Similarly, no detrimental effect of being on a very-low-protein diet was recorded in older patients in the “pre-dialysis” phase [38]. This is consistent with the results of our study.

However, in previous studies, the selection criteria usually included a heterogeneous population of patients at different levels of malnutrition risk (low, moderate, and high), and body composition was not evaluated, while our nutritional approach was tested in a cohort of homogeneous older adults with no nutritional risk status at baseline. Interestingly, according to the GNRI, an LPAE diet does not seem to raise the incidence of malnutrition risk, while it improves body composition by increasing FFM and skeletal muscle mass and decreasing FM and WC after 6 months on the LPAE diet. On the other hand, an increased incidence of malnutrition risk (8.7%) was observed in the non-compliant group. The obtained improvement in body composition and decrease in WC represent a novelty compared to previous studies, which, on the contrary, a worsening was observed in anthropometric and body composition parameters [39,40]. In particular, Barril and colleagues evaluated the effects of a protein intake < 0.8 g/Kg of IBW/day on body composition in CKD stage 3–5 patients [41]. They concluded that the subjects with the lowest protein intake experienced a worsening in body composition with a loss of muscle mass; no data, however, were provided regarding the energy intake of these patients.

Furthermore, we observed no significant difference in mean albumin, hemoglobin, body weight, or BMI values in either the compliant or non-compliant group. It is important to note that energy intake is of crucial importance in the dietary management of CKD elderly patients when dietary protein restriction is prescribed. In fact, while, in the compliant group, the protein intake should be sufficient to preserve muscle mass and nutritional status, in the non-compliant group, since the energy intake is adequate, a lower energy and a higher protein intake compared to the amount prescribed directly cause an increased malnutrition risk. Therefore, we demonstrate that an appropriate low-protein diet with adequate energy intake is nutritionally safe in order to improve body composition without increasing malnutrition risk, thus reconciling low protein intake with a good nutritional status in CKD elderly patients. Further studies are required to investigate the long-term efficacy and feasibility of the LPAE diet in preventing malnutrition risks in CKD elderly patients.

The secondary objectives of the study were to evaluate whether an LPAE diet could have a specific indication in the metabolic complications of CKD.

The systematic review by Rhee et al., in agreement with our data, concluded that low-protein diets (<0.8 g/Kg of IBW/day) were associated with an improvement in CKD metabolic abnormalities, not leading to protein-energy malnutrition or safety concerns [42]. However, in these previous studies, dietary energy intake recommendations were not always advised [42]. Notably, our study showed important opposite changes in the metabolic parameters of urea, phosphorus, uric acid, PTH levels, and the TG/HDL ratio, which are key targets in the conservative management of CKD, in both the compliant and non-compliant groups.

The pathological effects of phosphate on bones and the cardiovascular system motivate interventions to treat hyperphosphatemia. In clinical practice, the interventions to reduce phosphate levels consist of dietetic restrictions and phosphate binders. Although dietary phosphorus management is complex, there is a close relationship between protein and phosphorus intake [43]. Proteins are rich in phosphorus, so most of the scientific societies recommend reducing protein intake from the early stages in patients with CKD, to reduce the phosphorus intake. One gram of protein contains 13–15 mg of phosphorus, of which 30–70% is absorbed through the intestine [44]. Thus, an intake of 90 g of protein a day results in the absorption of 600–700 mg of phosphorus daily. Previous studies suggest that following a low-protein diet with a reduced phosphorus content lowers serum phosphorus, improves secondary hyperparathyroidism, and may delay kidney function decline [45,46,47]. Further experimental and clinical studies have shown that a low-phosphorus diet prevents the secondary hyperparathyroidism and hyperphosphatemia associated with CKD [48]. As such, an 800–1000 mg/d phosphorus recommendation has become a foundational part of the nutritional guidelines in CKD for individuals in stage 3–5 with serum phosphorus or PTH above a target level [21]. According to previous studies, reducing protein intake following the LPAE diet lowered phosphate and PTH serum levels, while they worsened in the non-compliant group, with a significant percentage change between groups (Figure 1).

Moreover, in addition to the increased phosphate levels observed in CKD patients, there is growing evidence that serum urea is a toxin, and there is no pharmacological therapy to treat and reduce its levels [49]. The reduction in several molecules derived by cellular metabolism, such as urea, obtained with low-protein diets has a positive impact on inflammatory status and pro-thrombotic events, with the consequent effect of reducing cardiovascular risk in CKD patients [50]. Additionally, urea is a contributing factor to insulin resistance, as circulating urea directly impairs insulin secretion by pancreatic islets [51]. As expected, the urea serum levels lowered as a consequence of the adherence to the LPAE diet. Similarly, uric acid levels also decreased in the compliant group after six months on the LPAE diet, with a significant percentage change between groups (Figure 1). Hence, multiple clinical and experimental data suggest that uric acid exerts pro-oxidant actions, inducing an increased production of oxygen free radicals that is potentially harmful to the vascular wall and other tissues [52]. Furthermore, uric acid is another factor that can contribute to IR, since it blocks the insulin-mediated endothelial nitric oxide release that is critical to insulin action [53]. Therefore, IR and uric acid are independently linked, and IR in metabolic syndrome models can be improved by lowering the serum uric acid [54]. In our CKD compliant patients, the LPAE diet directly lowered uric acid and urea levels, likely improving the insulin resistance as a consequence. The TG/HDL-C ratio may be the best reliable marker for predicting IR in the non-obese elderly population [55]. Indeed, the observed decrease in TG/HDL-C ratio in the compliant group, after 6 months on the LPAE diet, confirms the improvement in insulin sensitivity proposed by Rigalleau et al., who claimed that a low-protein diet played an effective beneficial role against insulin resistance independently of its influence on the progression of CKD [56]. Remarkably, when data were expressed as the percentage change from baseline in each group, significant differences between the compliant and non-compliant groups were also observed for the TG/HDL-C ratio (Figure 1). In particular, compliant patients experienced a statistically significantly greater benefit in terms of reduced uric acid and a reduced TG/HDL-C ratio compared to non-compliant patients, suggesting that the LPAE diet is an effective dietary strategy for improving metabolic control in CKD patients. Notably, the decreased abdominal fat defined by WC may be also a responsible factor for the improvement in insulin sensitivity in the compliant group (Table 2). All of this is very important during aging, as skeletal muscle dysfunction synergizes with visceral fat accumulation, with both amplifying the IR that occurs even in the very early stages of CKD.

Moreover, a low-protein diet, using protein-free products, may help to reduce sodium intake. Bellizzi et al. showed a significant reduction in sodium intake (with a reduction of 27% in the excreted fraction of sodium) and blood pressure levels after six months on a very-low-protein diet compared to any variation in a low-protein diet [57]. However, in our study, another notable finding regarding the benefits of the LPAE diet, although not the focus of our study, is that we found no significant blood pressure lowering effects in the compliant group, while significantly increased blood pressure was observed in the non-compliant group after 6 months on the LPAE diet.

The major strength of this study is that the nutritional status of the patients was comprehensively studied according to a number of common biochemical and clinical criteria or other tools for the assessment of malnutrition risk in CKD. A limitation to the study is that this is an observational, but not randomized, trial.

## 5. Conclusions

This observational study aims to answer the long-standing question of the nutritional management of elderly CKD patients, seeking to alleviate concerns regarding a lower protein intake in these subjects. In fact, in CKD elderly subjects, a LPAE diet does not imply a risk of developing malnutrition; rather, it was proven to improve the nutritional status and body composition. Notably, the LPAE diet increased the FFM and decreased the FM and WC—the latter a warning sign for visceral fat—and contributed to improving CKD metabolic abnormalities, including IR. Further longitudinal randomized studies on a larger population sample will certainly be needed to confirm the long-term nutritional safety of the LPAE diet in elderly patients.

## Figures and Tables

**Figure 1 nutrients-16-00632-f001:**
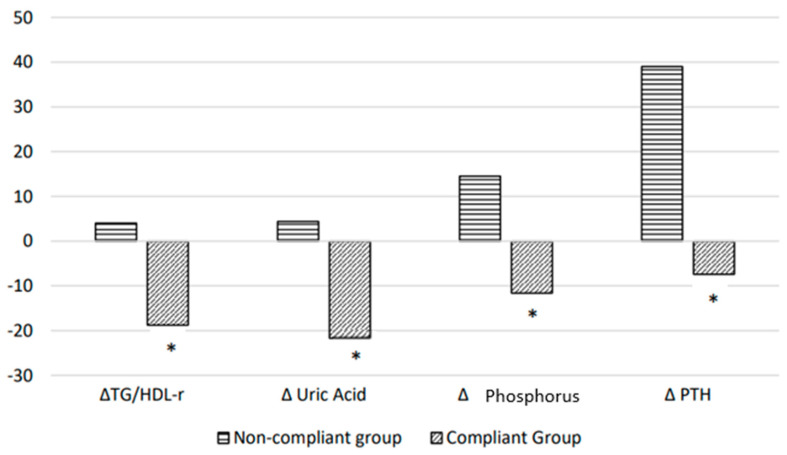
Percentage change in TG/HDL-ratio, blood uric acid, blood phosphorus, and PTH from compliant and non-compliant patients with CKD, before and after 6 months on the LPAE diet. * *p* values < 0.05 vs. Non-Compliant group. Abbreviations: TG/HDL ratio, triglyceride/high-density lipoprotein ratio; PTH, parathyroid hormone.

**Table 1 nutrients-16-00632-t001:** Anthropometric features, body composition characteristics, and metabolic parameters of the study population.

Baseline Anthropometric Features, Body Composition Characteristics, and Metabolic Parameters	
Male sex, n (%)	35 (83.3%)
Age, years	71.5 ± 5.5
Weight, Kg	74.7 ± 13.4
BMI, Kg/m^2^	27.3 ± 4.1
Waist circumference (WC), cm	97.7 ± 11.9
Fat-free mass, %	75.9 ± 7.9
Skeletal muscle mass, Kg	38.6 ± 4.6
Skeletal mass index, Kg/m²	14.1 ± 1.3
Fat mass,%	24.2 ± 8.1
Phase angle, Φ	5.4 ± 1.1
Systolic blood arterial pressure, mmHg	131.2 ± 14.1
Diastolic blood arterial pressure, mmHg	77.4 ± 9.7
Hypertension (n, %)	n. 37 (88.1%)
Dyslipidemia (n, %)	n. 21 (50.0%)
Glomerular filtration rate (mL/min/1.73 m²)	25.8 ± 11.5
CKD stage	Stage 3 (N.15; 35.7%) Stage 4 (N 19; 45.2%) Stage 5 (N 8; 19.0%)
Creatinine, mg/dL	2.9 ± 1.2
Blood urea, mg/dL	95.3 ± 38.3
Potassium, mg/dL	4.9 ± 0.6
Phosphorus, mg/dL	3.8 ± 1.0
Total calcium, mg/dL	9.4 ± 0.5
Albumin, g/dL	4.2 ± 0.4
Total cholesterol, mg/dL	165.9 ± 37.3
HDL cholesterol, mg/dL	43.3 ± 11.8
Triglycerides, mg/dL	138.0 ± 73.9
Hemoglobin, g/dL	12.9 ± 1.7
Glucose, mg/dL	93.4 ± 13.2
Uric acid, mg/dL	7.0 ± 1.6
PTH, pg/mL	147.9 ± 120.9
TG/HDL ratio	3.6 ± 2.5

Abbreviations: GNRI, geriatric nutritional risk index; BMI, body mass index, HDL, high-density lipoprotein; TG/HDL ratio, triglyceride/high-density lipoprotein ratio; PTH, parathyroid hormone.

**Table 2 nutrients-16-00632-t002:** Dietary features of the compliant and non-compliant groups at the baseline and after 6 months on the LPAE diet.

	Compliant n. 19, 45.2%		Non-Compliant n. 23, 54.8%	
	Prescribed	After 6 Months	Prescribed	After 6 Months
Protein Intake (g/IBW/day)	0.7 ± 0.1	0.7 ± 0.1	0.7 ± 0.1	0.8 ± 0.1 *
Energy Intake (kcal/IBW/day)	29.0 ± 3.2	28.8 ± 3.1	30.7 ± 3.1	25.6 ± 3.4 *

** p* values < 0.05 vs. baseline. Abbreviations are: IBW, ideal body weight.

**Table 3 nutrients-16-00632-t003:** Anthropometric parameters, malnutrition risk (GNRI) and the body composition features of compliant and non-compliant patients at baseline and after 6 months on the LPAE diet.

	Compliant Group n. 19		Non-Compliant Group n. 23	
	Baseline	After 6 Months	Baseline	After 6 Months
Weight, Kg	77.4 ± 15.7	75.7 ± 14.5	72.5 ± 11.0	71.3 ± 8.8
BMI, Kg/m^2^	27.8 ± 4.1	27.2 ± 3.7	26.9 ± 4.1	26.5 ± 3.3
Waist circumference, cm	101.1 ± 10.3	98.4 ± 9.1 *	95.4 ± 12.7	94.3 ± 11.6
GNRI ≥ 98, n (%)	19 (100%)	19 (100%)	23 (100%)	21 (91.3%)
Fat-free mass, %	77.1 ± 7.5	77.8 ± 7.1 *	76.5 ± 8.1	76.9 ± 9.3
Skeletal mass, Kg	39.2 ± 46	40.5 ± 5.5 *	37.9 ± 4.6	36.9 ± 5.1
Skeletal mass index, Kg/m²	14.2 ± 1.1	14.6 ± 1.2 *	14.1 ± 1.4	13.7 ± 1.4
Fat mass, %	23.4 ± 8.1	21.8 ± 7.6 *	23.5 ± 8.1	23.0 ± 9.3
Phase angle, Φ	5,3 ± 0.9	5.4 ± 0.8	5.6 ± 1.1	5.7 ± 1.4
Systolic blood arterial pressure, mmHg	134.2 ± 14.5	132.9 ± 15.1	128.9 ± 13.6	136.4 ± 13.0 *
Diastolic blood arterial pressure, mmHg	79.1 ± 11.1	76.1 ± 11.7	76.0 ± 8.5	79.5 ± 9.8

** p* values < 0.05 vs. baseline. Abbreviations: GNRI, Geriatric Nutritional Risk Index; BMI, body mass index.

**Table 4 nutrients-16-00632-t004:** Metabolic features of the compliant and non-compliant groups at baseline and after 6 months.

	Compliant n. 19		Non-Compliant n. 23	
	Baseline	After 6 Months	Baseline	After 6 Months
Blood glucose, mg/dL	92.8 ± 12.8	85.6 ± 13.1 *	94.4 ± 13.8	96.4 ± 26.8
Total cholesterol, mg/dL	162.8 ± 43.2	160.0 ± 35.1	168.4 ± 32.5	179.6 ± 36.8
HDL cholesterol, mg/dL	38.4 ± 9.6	41.4 ± 8.7 *	47.4 ± 12.1	47.4 ± 12.5
Triglycerides, mg/dL	141.9 ± 73.4	111.9 ± 40.2 *	134.8 ± 75.9	128.6 ± 51.8
Creatinine, mg/dL	2.8 ± 1.2	2.8 ± 1.4	2.9 ± 1.2	3.2 ± 1.6
Blood urea, mg/dL	96.7 ± 36.9	71.4 ± 28.8 *	94.2 ± 40.1	86.6 ± 39.1
Uric acid, mg/dL	7.7 ± 1.6	5.8 ± 1.4 *	6.5 ± 1.5	6.6 ± 1.5
Phosphorus, mg/dL	4.2 ± 1.1	3.5 ± 0.7 *	3.4 ± 0.6	3.9 ± 0.7 *
Potassium, mg/dL	4.8 ± 0.6	4.8 ± 0.4	5.0 ± 0.6	5.1 ± 0.6
Calcium, mg/dL	9.5 ± 0.7	9.4 ± 0.6	9.2 ± 0.3	9.4 ± 0.6
PTH, pg/mL	140.8 ± 132.5	102.4 ± 115.9 *	153.4 ± 113.9	206.1 ± 168.4 *
eGFR (CKD-EPI), mL/min/1.73 m^2^	26.7 ± 10.1	28.5 ± 14.1	25.4 ± 12.7	24.4 ± 12.4
TG/HDL ratio	3.9 ± 2.4	2.8 ± 1.3 *	3.3 ± 2.6	3.1 ± 1.8
Hemoglobin, g/dL	13.2 ± 1.9	13.0 ± 1.7	12.7 ± 1.5	12.4 ± 1.6
Albumin, g/dL	4.2 ± 0.5	4.0 ± 0.3	4.1 ± 0.3	4.1 ± 0.4

** p* values < 0.05 vs. baseline. Abbreviations: GNRI, Geriatric Nutritional Risk Index; BMI, body mass index, HDL high-density lipoprotein; TG/HDL ratio, triglyceride/high-density lipoprotein ratio; PTH, parathyroid hormone.

## Data Availability

The data are stored in a database at the Department of Clinical Medicine and Surgery, Nutrition Physiology Unit, University Federico II of Naples, Naples 80131, Italy. It is available upon request to be made to Bruna Guida who a co-authors of the paper.

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
