# Peer review of "Effects of Dietary Intervention on Nutritional Status in Elderly Individuals with Chronic Kidney Disease"

_nutrients, 2024, doi:10.3390/nu16050632_

Round 1

Reviewer 1 Report

Comments and Suggestions for Authors

The manuscript “Effects of Dietary Intervention on Nutritional Status in the Elderly with Chronic Kidney Disease” investigated the dietary intervention on nutritional status in the elderly with CKD. The authors have given some interesting results and some revisions should be modified as follows:

1. In Line 18, “To this purpose”, there should be a comma after this sentence.

2. In the section introduction, did the low-protein adequate energy intake (LPAE) diet apply in other studies to alleviate CKD? The authors needed to introduce the benefits of the LPAE and add some references to support its merit.

3. In Line 82: this study lacks ethical proof, authors should add the number of ethical proofs in this study.

4. In Lines 105-108, the methods of determining biochemical parameters were lacking, authors can refer to https://doi.org/10.1007/s00784-023-04989-1 and https://doi.org/10.1016/j.scitotenv.2023.164808.

5. In the section of the discussion, the authors needed to describe the previous studies of LPAE, and compare their results to previous studies.

6. In Line 235, “in this study” should have a comma after this sentence.

7. Line 240, “In 2022” should have a comma after this sentence.

8. There are many short paragraphs in the discussion section, which the author can combine moderately.

9. In the section of in conclusion, authors needed to directly point out the novelty of their studies.

10. The defects of this paper or further studies can also be reflected in the conclusion.

11. Please carefully check the format of references.

Reviewer 2 Report

Comments and Suggestions for Authors

Dear Editor and Authors,

With great interest, I have read the manuscript " Effects of Dietary Intervention on Nutritional Status in the Elderly with Chronic Kidney Disease". This is an interesting topic and the results of this study bring novel findings to the field. Before it can be proceeded I suggest some corrections:

Introduction: It is well written, explains the main problem of the topic

Materials and methods: This section is well written. The authors have explained in detail the whole process of the study.

Results: This section needs to be rewritten clearly and fluently. Authors in the beginning mentioned all tables. Results should be rewritten from the beginning, acknowledging the number of patients, their demographic data, etc, and then explaining the main results from each table with P values. It is a little bit confusing.

Discussion: This section is well written, same as the Conclusion

References: They should be rewritten according to journal guidelines.

Comments on the Quality of English Language

Minor editing of English language is required.

Round 2

Reviewer 1 Report

Comments and Suggestions for Authors

It can be accepted in the present form.